# Baicalin Attenuates H_2_O_2_-Induced Oxidative Stress by Regulating the AMPK/Nrf2 Signaling Pathway in IPEC-J2 Cells

**DOI:** 10.3390/ijms24119435

**Published:** 2023-05-29

**Authors:** Jiahua Liang, Ying Zhou, Xinyi Cheng, Jiaqi Chen, Huabin Cao, Xiaoquan Guo, Caiying Zhang, Yu Zhuang, Guoliang Hu

**Affiliations:** Jiangxi Provincial Key Laboratory for Animal Health, Institute of Animal Population Health, College of Animal Science and Technology, Jiangxi Agricultural University, No. 1101 Zhimin Avenue, Economic and Technological Development District, Nanchang 330045, China

**Keywords:** baicalin, oxidative stress, IPEC-J2 cell, AMPK-mediated antioxidant defense, apoptosis

## Abstract

Oxidative stress can adversely affect the health status of the body, more specifically by causing intestinal damage by disrupting the permeability of the intestinal barrier. This is closely related to intestinal epithelial cell apoptosis caused by the mass production of reactive oxygen species (ROS). Baicalin (Bai) is a major active ingredient in Chinese traditional herbal medicine that has antioxidant, anti-inflammatory, and anti-cancer properties. The purpose of this study was to explore the underlying mechanisms by which Bai protects against hydrogen peroxide (H_2_O_2_)-induced intestinal injury in vitro. Our results indicated that H_2_O_2_ treatment caused injury to IPEC-J2 cells, resulting in their apoptosis. However, Bai treatment attenuated H_2_O_2_-induced IPEC-J2 cell damage by up-regulating the mRNA and protein expression of ZO-1, Occludin, and Claudin1. Besides, Bai treatment prevented H_2_O_2_-induced ROS and MDA production and increased the activities of antioxidant enzymes (SOD, CAT, and GSH-PX). Moreover, Bai treatment also attenuated H_2_O_2_-induced apoptosis in IPEC-J2 cells by down-regulating the mRNA expression of Caspase-3 and Caspase-9 and up-regulating the mRNA expression of FAS and Bax, which are involved in the inhibition of mitochondrial pathways. The expression of Nrf2 increased after treatment with H_2_O_2_, and Bai can alleviate this phenomenon. Meanwhile, Bai down-regulated the ratio of phosphorylated AMPK to unphosphorylated AMPK, which is indicative of the mRNA abundance of antioxidant-related genes. In addition, knockdown of AMPK by short-hairpin RNA (shRNA) significantly reduced the protein levels of AMPK and Nrf2, increased the percentage of apoptotic cells, and abrogated Bai-mediated protection against oxidative stress. Collectively, our results indicated that Bai attenuated H_2_O_2_-induced cell injury and apoptosis in IPEC-J2 cells through improving the antioxidant capacity through the inhibition of the oxidative stress-mediated AMPK/Nrf2 signaling pathway.

## 1. Introduction

The intestinal epithelium, as an important component of the intestinal physical barrier, plays a key role in absorbing nutrients and protecting organisms from harmful substances such as antigens and pathogens. The intestinal barrier consists of a layer of epithelial cells connected by tight junction proteins (TJs), such as Zonula, Occludin, and Claudins. Various harmful factors, such as toxins and pathogens that constantly interact with the intestinal epithelium in the intestinal tract, can induce the overproduction of free radicals, which trigger an inflammatory response and oxidative stress [1]. In addition, the disruption of the intestinal barrier will increase intestinal permeability and bacterial infiltration and trigger systemic inflammation, leading to the appearance of intestinal diseases, such as inflammatory bowel disease and colon cancer [2]. Recently, studies have revealed that oxidative stress induced by reactive oxygen species is one of the main factors causing several cellular abnormalities, such as DNA damage and cell death, which have been reported to correlate with intestinal disease through the excessive accumulation of reactive oxygen species (ROS) [3,4,5].

Oxidative stress is defined as the process by which an organism produces excess reactive oxygen species and free radicals in the presence of certain stimuli, which could cause physiological and pathological reactions in tissues and cells. The production and removal of free radicals in organisms occur in a dynamic equilibrium state; while free radicals are produced excessively and cannot be cleared in time, they could cause cell damage (via lipid peroxidation, abnormal protein folding, mitochondrial dysfunction, and apoptosis) [6,7]. Recent studies have shown that oxidative stress induced by ROS could disturb cellular functions (disruption of mitochondrial function and blocking autophagy) by influencing the redox-sensitive pathway and transcription factors/activators, such as nuclear factor erythroid-2-related factor 2 (Nrf2) and the AMP-activated protein kinase (AMPK) signaling pathway [8]. Nrf2 is a redox-sensitive transcription factor that inhibits oxidative stress production by regulating the expression of antioxidant enzyme genes, and AMPK is an important energy metabolism signal pathway gene that affects mitochondrial function and autophagy. AMPK and Nrf2 have an important regulatory effect on oxidative stress and synergize with other transcription factors to produce an antioxidant and anti-inflammatory effect [9,10]. In addition, activating AMPK induced mitochondrial autophagy, which reduced ROS overproduction, intestinal epithelium barrier injury, and mitochondrial dysfunction [11].

As a plant extract, Baicalin (Bai) is an active flavonoid compound originating and extracted from the root tubers of *Scutellaria baicalensis* Georgi, which has anti-inflammatory, anti-oxidation, and anti-tumor properties [12,13]. Kong and his colleagues showed that Bai can inhibit the proliferation of cancer cells and induce apoptosis and autophagy both in vitro and in vivo [14]. A study showed that Bai could decrease the expression of pro-inflammatory cytokines and increase the antioxidant capacity in the serum and intestinal tissue, which attenuated intestinal inflammatory and oxidative damage caused by deoxynivalenol [15]. In addition, Bai acts on multiple cellular targets such as AMPK, protein kinase B (Akt), nuclear factor Kappa B (NF-κB), and Nrf2 to control processes and signaling pathways related to reduction of ROS production and apoptosis [16,17]. Based on previous studies, Bai is known to have substance-reported maintenance of mitochondrial function and autophagy, oxidative stress-reducing effects, and intestinal barrier strengthening effects; therefore, it has become an interesting research target for prevention strategies to reduce intestinal injury induced by oxidative stress. However, the target genes and mechanisms of Bai in alleviating oxidative stress remain unclear. To explore whether the AMPK and Nrf2 pathways participate in Bai preventing H_2_O_2_-induced intestinal barrier damages, we determined antioxidant enzyme activity and relative gene expression of apoptosis in the IPEC-J2 cells that were pretreated with a series of Bai concentrations.

## 2. Results

### 2.1. Bai Attenuated H_2_O_2_-Induced Cell Damage and Elevated the Intestinal Barrier Function of IPEC-J2 Cells

To investigate the cytotoxicity of Bai on IPEC-J2 cells, the cell viability was tested by CCK-8 assays, and the morphological changes of IPEC-J2 cells were observed by optical microscopy. As shown in Figure 1A,B, the viability of IPEC-J2 cells treated with H_2_O_2_ was significantly decreased compared with control group (0 μM H_2_O_2_). However, pretreatment of IPEC-J2 cells with Bai significantly improved cell viability in the presence of H_2_O_2_ in a dose-dependent manner, which was consistent with morphological change by light microscope observation. To further determinate the effects of Bai on H_2_O_2_-induced barrier function damage in IPEC-J2 cells, we detected the TJ protein expression (Figure 1E,G). Compared with the control group, the mRNA abundance of ZO-1 and Occludin was significantly decreased in the H_2_O_2_-treated group. Pretreating with Bai reversed this downregulation of Occludin induced by H_2_O_2_ (*p* < 0.05). The protein expression of Claudin-1, ZO-1, and Occludin was decreased in the H_2_O_2_-treated group compared with the control group, while the addition of Bai can effectively alleviate this decline. In addition, immunofluorescence results of Occludin were consistent with TJ protein expression (Figure 1C,D).

### 2.2. Bai Attenuated H_2_O_2_-Induced Apoptosis in IPEC-J2 Cells

Apoptosis occurs as a result of the H_2_O_2_-exposed cytopathic effect and cell death. To investigate the protective effect of Bai on H_2_O_2_-induced IPEC-J2 cell apoptosis, the apoptotic cells were assayed by flow cytometry after PI and annexin V-FITC staining. The apoptosis rate of IPEC-J2 cells in H_2_O_2_ treatment was significantly increased compared with the control group (*p* < 0.001). However, pretreatment with Bai significantly suppressed cell apoptosis induced by H_2_O_2_ exposure (Figure 2A,B). To further confirm the effects of Bai on the apoptotic pathways in H_2_O_2_-induced IPEC-J2 cells, we detected the mitochondrial apoptosis pathway in protein levels. As shown in Figure 2C,D, western blot analysis showed that Bai treatment prevented the expression of pro-apoptotic proteins induced by H_2_O_2_ (*p* < 0.05 or *p* < 0.01), whereas no significant changes were found in the protein levels of Bcl2 and Caspase9 (*p* > 0.05).

### 2.3. Bai Attenuated H_2_O_2_-Induced ROS Production through Improved Antioxidant Enzyme Activities and Activated the AMPK Signaling Pathway in IPEC-J2 Cells

To confirm whether H_2_O_2_-induced oxidative stress was attenuated by Bai increasing the antioxidant enzyme activities, the intracellular ROS level and activation of antioxidant capacity were detected in IPEC-J2 cells. Compared with the control group, the intracellular ROS level increased significantly in the H_2_O_2_-treated group, whereas Bai reversed the H_2_O_2_-induced upregulation of ROS level (*p* < 0.01) (Figure 3A,B). The increase in intracellular ROS level depends on the breakdown of oxidative balance. As shown in Figure 3C, the antioxidant enzyme activities of SOD, CAT, and GSH-px were significantly decreased, and the lipid peroxidation products (MDA) were significantly increased in the H_2_O_2_-treated group, while pretreatment with Bai in a dose-dependent manner significantly increased SOD, CAT, and GSH-px activation and decreased MDA activation induced by H_2_O_2_ (*p* < 0.05 or *p* < 0.001). Antioxidant gene expression was regulated by the AMPK/Nrf2 signaling pathway. As shown in Figure 3D,E, the antioxidant-related genes SOD, GPX1, Keap1, and SESN2 were significantly increased in the H_2_O_2_-treated group, while pretreatment with Bai significantly reversed this upregulation of SOD, GPX1, Keap1, and SESN2 induced by H_2_O_2_ (*p* < 0.001 or *p* < 0.05). In addition, H_2_O_2_ increased the ratio of p-AMPK/AMPK and decreased the ratio of p-Nrf2/Nrf2 compared with the control group, which was reversed with the addition of Bai (*p* < 0.05) (Figure 3F,G).

### 2.4. Knockdown of AMPK Blocked the Protective Effect of Bai on H_2_O_2_-Induced Intestinal Barrier Damage and Apoptosis

To further investigate whether Bai protects IPEC-J2 cells against H_2_O_2_-induced intestinal epithelial cell damage via the AMPK signaling pathway, we silenced AMPK gene expression by shRNA in IPEC-J2 cells and then treated the AMPK-knockdown cells with Bai and/or H_2_O_2_. The TJ protein expression and cell apoptosis were determined in AMPK-knockdown IPEC-J2 cells and normal (SC) IPEC-J2 cells. As shown in Figure 4A–D, knockdown of AMPK by shRNA has little effect on cell viability and TJ protein expression according to the results of Occludin immunofluorescence and Claudin1 mRNA expression. However, knockdown of AMPK in IPEC-J2 cells largely abrogated the beneficial effects of Bai pretreatment on the H_2_O_2_-induced decrease in cell apoptosis and the ratio of Bcl2/BAX mRNA levels (Figure 4E–G).

### 2.5. Knockdown of AMPK Blocked the Protective Effect of Bai on H_2_O_2_-Induced Oxidative Stress

To confirm whether Bai functioned through the AMPK/Nrf2 signaling pathway to prevent H_2_O_2_-induced oxidative stress, we tested the ROS level, measured the mRNA expression of antioxidant genes (SOD, CAT, and GPX), and determined protein levels of AMPK and Nrf2. As shown in Figure 5A,B, knockdown of AMPK by shRNA has significantly increased the ROS level in IPEC-J2 cells induced by H_2_O_2_ (*p* < 0.01). However, pretreatment with Bai has little effect on the change in ROS level in AMPK-silenced IPEC-J2 cells. The mRNA expression and heatmap analysis of antioxidant gene analysis showed that knockdown of AMPK downregulates the increased antioxidant gene expression induced by H_2_O_2_ (*p* < 0.05). However, pretreatment with Bai showed no significant difference compared with the H_2_O_2_ group in AMPK-knockdown IPEC-J2 cells (Figure 5C–F). In addition, H_2_O_2_ treatment significantly decreased the protein levels of AMPK and Nrf2 in AMPK-knockdown cells compared with the SC group (*p* < 0.05) (Figure 5G–I).

## 3. Discussion

The intestine is not only one of the major digestive and absorptive organs but also an innate barrier that protects the organism from various infectious agents and toxins. Numerous studies have shown that the occurrence and development of clinical diseases such as inflammatory bowel disease (IBD) and fatty liver disease are closely related to intestinal barrier damage [18,19]. The accumulation of active oxygen will cause molecular biological damage, which is a critical factor involved in disruption of the intestinal barrier, leading to intestinal diseases [20,21,22]. Oxidative stress is also the cause of intestinal barrier damage, and the identification of appropriate plant extracts with known antioxidant activity as therapeutic agents to counteract oxidative stress-induced intestinal injury is a research hotspot at present. In the present study, our results showed that Bai significantly attenuated H_2_O_2_-induced IPEC-J2 cell injury, which could increase cell viability, upregulate the ability of antioxidant enzymes, and regulate tight junction protein levels. Besides, Bai, through activation of the AMPK signaling pathway, regulated apoptosis and protein synthesis to exert cytoprotective activity.

As a dietary supplement, Baicalin, a flavonoid extracted from a variety of Chinese herbs with a long history of clinical application, has been identified to have multiple biological functions. Bai was identified to have antioxidant, antibacterial, anticancer, anti-aging skin, etc. properties, and thus, it has received great attention due to its potential medicinal value [23]. Accumulated evidence has indicated that Bai could protect from intestinal injury by improving intestinal barrier function and antioxidation activity [24]. In the present study, our results showed that Bai, in a low dose, significantly up-regulated the expression of genes encoding the tight junction protein in IPEC-J2 cells, suggesting that Bai had a certain application value in protecting against H_2_O_2_ injury. The genes encoding the tight junction protein (Occludin, Claudins, and ZO-1) are the principal determinants of intestinal barrier integrity [25]. These tight junction proteins, through strengthening intestinal epithelial junctions, resist invasion by harmful substances and reduce stress damage, thereby exerting a protective function in the intestinal barrier. A study showed that the phenazine biosynthesis-like domain-containing protein (PBLD) could inhibit pro-inflammatory cytokine production by blocking NF-κB activation, reduce the production of inflammatory mediators, and protect colon barrier integrity via enhancing tight junction protein [26]. Previous research clarified in vitro the mechanism by which Bai transforms into baicalein under the action of intestinal flora, and the baicalein can protect intestinal villi, prevent intestinal diseases, and enhance anti-stress damage abilities [27]. Cheng and his colleagues showed that through the NF-κB signal pathway, Bai can protect APEC-induced intestinal injury, increase the expression of intestinal barrier protein, alleviate intestinal stress, and decrease intestinal villus [28]. Risha Ganguly’s study showed that Bai could eliminate the formation of free radicals and protect against fluoxetine-induced hepatotoxicity and liver injury by regulating oxidative stress and inflammation [29]. H_2_O_2_ is one of the primary sources of ROS and oxidative stress damage to the cells, and the aim of our study was to determine the mechanisms of oxidative stress on cell injury, which has important relevance to our current understanding of disorders.

Oxidative stress was described as a state occurring as a result of an imbalance between the activity of free radicals such as ROS and reactive nitrogen species (RNS) and the antioxidative system in an organism. Recently, studies have revealed that oxidative stress is a common mechanism that destroys intestinal permeability, which induces inflammation that further aggravates and increases oxidative stress. H_2_O_2_, as an intracellular redox signaling molecule, is commonly used in studies of redox-regulated processes. It has been well documented that the exposure of cells to H_2_O_2_ triggers the overproduction of ROS both in vivo and in vitro [20]. In the present study, IPEC-J2 cells exposed to H_2_O_2_ showed sharply decreased cell viability, decreased expression of TJ proteins compared with the control group, and caspase-dependent mitochondrial pathway induced apoptosis. Accumulation of ROS and disturbance of innate antioxidant systems by exposure to H_2_O_2_ in a cell cause damage to barrier function and apoptosis in intestinal epithelial cells [30]. However, in the present study, pretreatment of cells with Bai significantly protected against H_2_O_2_-induced oxidative stress in IPEC-J2 cells by up-regulating the expression of TJ proteins and antioxidant genes and then sharply decreased ROS production and alleviation of apoptosis, as affirmed by activation of the Caspase-3/Bcl/Bax signal pathway. In addition, our results showed that Bai exhibited good antioxidant activity as determined through activities of antioxidant enzymes such as GSH-px, SOD, and CAT. GSH-Px and SOD avert oxidative stress through catalyzing glutathione oxidation and superoxiding anions to H_2_O_2_, and then CAT reduces redox damage by detoxifying H_2_O_2_ into H_2_O [31]. Besides, multiple studies have conceived that the MDA content is a reflection of lipid peroxidation damage when the cell suffers overproduction of ROS [32]. A study showed that Bai attenuated mycoplasma gallisepticum-induced oxidative stress through improving the activity of antioxidant enzymes and upregulating the expression of antioxidant genes and proteins [33]. Research has also shown that Bai could directly remove free radicals, superoxide anions, and other oxygen free radicals and inhibit the activity of xanthine oxidase. The AMPK signaling pathway can be activated by Bai, which improves mitochondrial function and regulates ROS levels. Li and colleagues revealed that Bai could regulate mitochondrial function and elevate the mitochondrial membrane potential via suppression of ROS production in a manner dependent on AMPK in vivo and in vitro [34].

AMPK, a sensor of cellular energy status that regulates cellular glucose and lipid metabolism, plays a pivotal role in regulating mitochondrial homeostasis and oxidative stress. AMPK and its orthologs exist in almost all eukaryotes, and their main function is to monitor the changes of intracellular energy, which combines adenosine-triphosphate (ATP) with phosphorylation of downstream substrates to regulate the rate of ATP-producing pathways [35]. Wu and colleagues showed that citrinin can induce liver injury, which activates the AMPK pathway in L02 cells, increases the phosphorylation of AMPK and decreases ATP content, and induces cell cycle arrest and apoptosis to aggravate liver injury [36]. Recently, studies revealed that the AMPK signaling pathway could regulate the transcription factor Nrf2, which is a pivotal modulator of the antioxidant defense response to oxidative stress and upregulate antioxidant gene (SOD, CAT, and GPX) expression. Moreover, Nrf2 was able to bind to the ARE binding site at AMPK which suggested that Nrf2 could directly regulate the activation of the AMPK signaling pathway. Previous studies have shown that down-regulation of AMPK could result in the inactivation of the Nrf2 signaling pathway and elevate ROS levels [37]. In the present study, our results showed that Bai could protect against oxidative stress-induced damage by elevating sestrin 2 (SESN2) and kelch-like ECH-associated protein 1 (Keap1) expression in the AMPK signaling pathway. Hu and colleagues showed that salidroside, a compound extracted from the dried roots and rhizomes of Rhodiola, can activate AMPK phosphorylation and inhibit NF-κB, p65, and NOD-like receptor thermal protein domain associated protein 3 (NLRP3) inflammatory bodies to reduce endothelial inflammation and oxidative stress; however, this protective effect of salidroside was abolished in the presence of compound C (an inhibitor of AMPK) [38]. A previous study found that Bai can activate the phosphorylation of inositol-requiring enzyme-1α (IRE1α) and AMPK expression to play an antioxidant role and reduce the accumulation of reactive oxygen species in the endoplasmic reticulum stress (ERS) response [39]. In addition, Bai can effectively inhibit the inflammation and apoptosis of interstitial cells through activation of the AMPK signaling pathway and decrease the expression of NF-κB to alleviate ulcerative colitis in rats [16]. In our study, the results demonstrated that the protective effect of Bai against H_2_O_2_-induced intestinal injury was abolished by shRNA interfering with AMPK expression in IPEC-J2 cells, which suggests that Bai protects against oxidative stress by activating the AMPK signaling pathway. Moreover, the phosphorylation level of Nrf2 was affected when AMPK was knocked down. These findings indicate that the AMPK pathway plays a pivotal role in the cytoprotective effects of Bai-induced AMPK activation against oxidative stress.

In summary, our results demonstrated that Bai could effectively protect IPEC-J2 cells against oxidative stress via AMPK activation and up-regulation of intestinal barrier proteins and promotion of antioxidant activity, thereby decreasing intracellular ROS levels and apoptosis. This study suggested that baicalin could be a potent and effective feed additive to protect against intestinal injury in livestock production.

## 4. Materials and Methods

### 4.1. Chemicals and Reagents

Bai were obtained from Selleckhem (Houston, TX, USA). Dulbecco’s modified Eagle’s medium (DMEM), fetal bovine serum (FBS), and antibiotics (penicillin and streptomycin) required for cell culture were obtained from Gibco (Carlsbad, CA, USA). The porcine intestinal epithelial cells (IPEC-J2 cells), the plasmids pLKO.1, pEGFP-C1, pCMV-DR8.9, and VSV-G were obtained from Fenghui (Changsha, China). The antibodies against GAPDH and p-Nrf2 were acquired from Bimake (Houston, TX, USA), the antibodies against Claudin1 and Occludin were from Selleckchem (Houston, TX, USA), the antibodies against AMPK, p-AMPK, and Nrf2 were from Wanlei Biotechnology Co., Ltd. (Shenyang, China), and the antibody against ZO-1 was from Proteintech (Wuhan, China).

### 4.2. Cell Culture and Treatments

The IPEC-J2 cell line and the 239T cell line were cultured in Dulbecco’s modified Eagle’s medium (DMEM) supplemented with 10% fetal bovine serum (FBS; Thermos Fisher Scientific, Waltham, MA, USA), and maintained at 37 °C in a humidified chamber of 5% CO_2_. According to experimental requirements, the cells were seeded in different areas of cell culture dishes (100-mm cell culture dish, 5 × 10^6^ cells/cm^2^; 6-well plate, 9 × 10^5^ cells/cm^2^; 96-well plate, 3 × 10^4^ cells/cm^2^). IPEC-J2 cells were pretreated with various doses of Bai (10 and 20 μM) for 12 h and then treated with H_2_O_2_ (500 μM) for 4 h.

### 4.3. Lentivirus Interfering Transfection

293T cells in each well were transfected with plasmids (3 μg, pLKO1: VSV-G: DR8.9 = 3:2:1) and pEGFP-C1 (0.2 μL/well). Three μL of Lipofectamine^TM^ 2000 (Thermos Fisher Scientific company, Waltham, MA, USA) transfection reagent was added to the cells, mixed, and incubated on ice for 20 min. After 12 h of incubation, the medium was replaced with DMEM containing 10% FBS per well, and the lentivirus venom was collected at the 48 h and 72 h. IPEC-J2 cells were cultured in 10% FBS at 37 °C in a 5% CO_2_ incubator and 1.8 μL of the lentivirus venom was added. After 12 h of culture, the puromycin (3 μL/mL) on the screening cells was replaced. The sequences for the different shRNAs are shown in Table 1.

### 4.4. Cell Viability Assay

Cell viability assays were performed by using a Cell Counting Kit-8 (CCK8) according to the manufacturer’s instructions. IPEC-J2 cells and shAMPK IPEC-J2 cells were seeded in a 12-well plate. When the IPEC-J2 cells density reached 1 × 10^6^ cells/well, they were treated with H_2_O_2_ (500 μM/well), ML385 (5 μM/well), and Bai (10 μM/well) for the indicated times. Following drug treatment, CCK8 (10 μL) was added to each well and incubated at 37 °C for another 2 h. The absorbance at 450 nm was measured on a microplate reader. All experiments were conducted independently in triplicate.

### 4.5. Determination of Oxidative Stress

After the various treatments, cell samples were centrifuged at 4000 r/min for 10 min at 4 °C to collect the samples. In addition, for IPEC-J2 cells, the freeze thaw method at −80 °C for three times was used to collect the supernatant. Superoxide dismutase (SOD) activity, catalase (CAT) activity, glutathione peroxidase (GSH-px), and malondialdehyde (MDA) levels were determined according to the manufacturer’s protocols (Nanjing Jiancheng Institute of Biotechnology, Nanjing, China).

### 4.6. Detection of Intracellular ROS Accumulation

A ROS kit (Beyotime Institute of Biotechnology, Shanghai, China) was used to detect the intracellular reactive oxygen level by flow cytometry. The activity was determined by the selective absorption of light by a solution.

### 4.7. Apoptosis Analysis

The treated IPEC-J2 cells were seeded in a 12-well plate and washed three times with PBS stored at 4 °C. The Annexin V-FITC/PI apoptosis detection kit (Beyotime Institute of Biotechnology, Shanghai, China) was used to further detect the apoptosis of IPEC-J2 cells according to the instruction manual, and the operation was completed within an hour. The apoptosis index is expressed as a ratio of the whole number of positive apoptotic cells.

### 4.8. Immunofluorescence Assay

The IPEC-J2 cells were fixed with 4% PFA for 30 min and rinsed three times in PBS. Triton X-100 (0.2%) was added to the fixed samples for 30 min and incubated in a blocking buffer (5% BSA) for 1 h. Then, cultured cells were stained with primary antibodies specific for Occludin (1:400 dilution) overnight at 4 °C. The cells were incubated with RBITC (1:1000 dilution) as secondary antibodies at 37 °C for 1 h in the dark. The cells were incubated with DAPI (2 μL/well) staining. Finally, specific fluorescence was imaged on a confocal microscope.

### 4.9. Real-Time Quantitative Polymerase Chain Reaction (RT-qPCR)

Total RNA was extracted from IPEC-J2 cells using Trizol (Takara, Kusatsu-shi, Janpan) reagent, and the concentration and purity of total RNA were measured by the Thermo NanoDrop 2000 (Thermo Fisher Scientific Inc., Waltham, MA, USA). Then, the RNA was reverse transcribed with the Fast Quant RT Kit (Takara, Kusatsu-shi, Japan). A SYBR Green PCR Master Mix (Novozan Biotechnology Co., Ltd., Nanjing, China) was used to perform quantitative real-time RT-qPCR. The PCR products were normalized to GAPDH mRNA levels. The relative mRNA transcriptional levels were calculated based on the 2^−ΔΔCT^ method. The primer sequences are presented in Table 2.

### 4.10. Western Blot

Cell samples were split using the RIPA assay (Solabao Technology company, Beijing, China) in ice. Total proteins were quantified using the BCA assay (Solabao Technology company, Beijing, China) and electrophoresed on 10% SDS-acrylamide gels. The blots were incubated with primary antibodies overnight at 4 °C. Blots were then incubated with an anti-rabbit or an anti-mouse secondary antibody for 40 min. Moreover, an anti-GAPDH antibody (1:5000) was employed as a loading control for protein expression. The protein bands were gauged on the BioRad Chemin Doc XRS system and quantified using Image J densitometry software.

### 4.11. Statistical Analysis

Statistical analysis was performed suing Microsoft Office Excel 2019 and SPSS 25.0 software. Differences between groups were analyzed by one-way analysis of variance (ANOVA) and unpaired t-tests. Pairwise comparisons were performed by LSD. Data are expressed as the mean ± SD and were considered statistically significant at *p* < 0.05, *p* < 0.01, and *p* < 0.001.

## Figures and Tables

**Figure 1 ijms-24-09435-f001:**
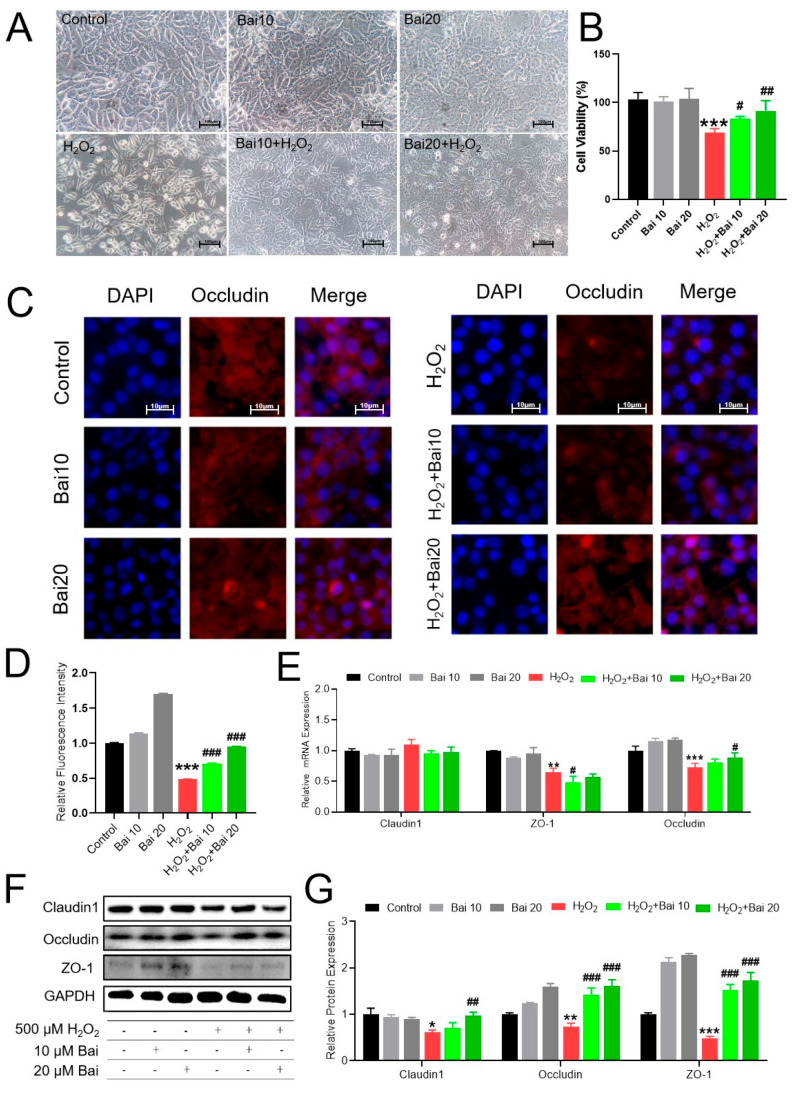
Effects of Baicalin (Bai) and H_2_O_2_ on the viability of IPEC-J2 cells and intestinal barrier damage. (**A**) Observation of the morphological changes in IPEC-J2 cells after treatment with Bai for 12 h and H_2_O_2_ for 4 h under an inverted microscope (200 total magnification). Scale bar, 100 μm. (**B**) CCK8 reagent measured the viability of IPEC-J2 cells. (**C**) Immunofluorescence was observed in the sections stained for Occludin under different conditions. Scale bar, 10 μm. (**D**) Quantitative analysis of Occludin relative fluorescence intensity. (**E**) The mRNA levels of intestinal barrier-related genes (ZO-1, Occludin, and Claudin1) in IPEC-J2 cells under different treatments. (**F**,**G**) Levels of intestinal barrier proteins. Data were represented as the mean ± SD of at least three independent experiments. “*” indicates a significant difference compared with the control group (* *p* < 0.05, ** *p* < 0.01, and *** *p* < 0.001). “#” indicates a significant difference compared with the H_2_O_2_ group (# *p* < 0.05, ## *p* < 0.01, and ### *p* < 0.001). Below is the same.

**Figure 2 ijms-24-09435-f002:**
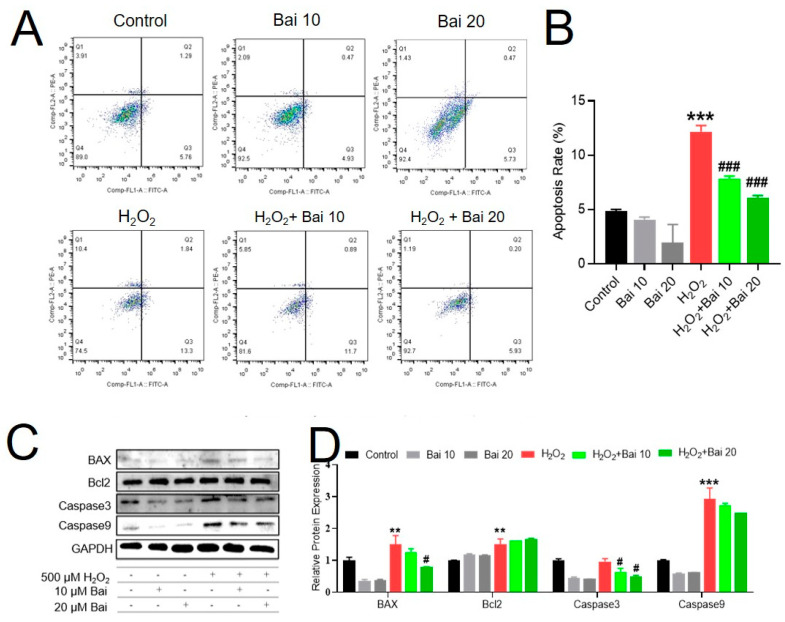
The protective effects of Bai on apoptosis in H_2_O_2_-induced IPEC-J2 cells. IPEC-J2 cells were pretreated with the indicated concentrations of Bai and then cocultured with 500 μM H_2_O_2_ for 4 h. (**A**,**B**) Apoptosis cells were analyzed by flow cytometry; results are presented as the percentage of cell viability; values are the mean ± SD. (**C**,**D**) Levels of apoptosis-related proteins. “*” indicates a significant difference compared with the control group (** *p* < 0.01 and *** *p* < 0.001). “#” indicates a significant difference compared with the H_2_O_2_ group (# *p* < 0.05 and ### *p* < 0.001).

**Figure 3 ijms-24-09435-f003:**
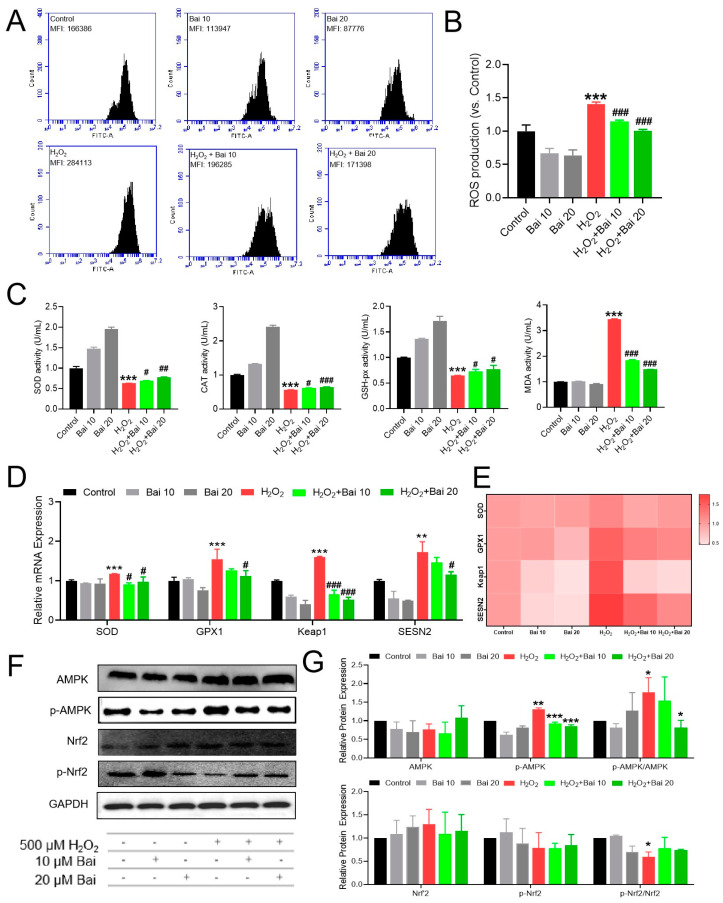
The protective effects of Bai on antioxidant oxidative stress markers in IPEC-J2 cells under H_2_O_2_ exposure. (**A**,**B**) ROS production; (**C**) SOD activity; CAT activity; GSH-px activity; MDA activity. (**D**,**E**) The mRNA levels of antioxidant oxidative stress-related genes (SOD, GPX1 Keap1, and SESN2). Heat map showing the expression levels of antioxidant oxidative stress-related mRNA expression in IPEC-J2 cells under different treatments. (**F**,**G**) Levels of antioxidant oxidative stress-related proteins. “*” indicates a significant difference compared with the control group (* *p* < 0.05, ** *p* < 0.01, and *** *p* < 0.001). “#” indicates a significant difference compared with the H_2_O_2_ group (# *p* < 0.05, ## *p* < 0.01, and ### *p* < 0.001).

**Figure 4 ijms-24-09435-f004:**
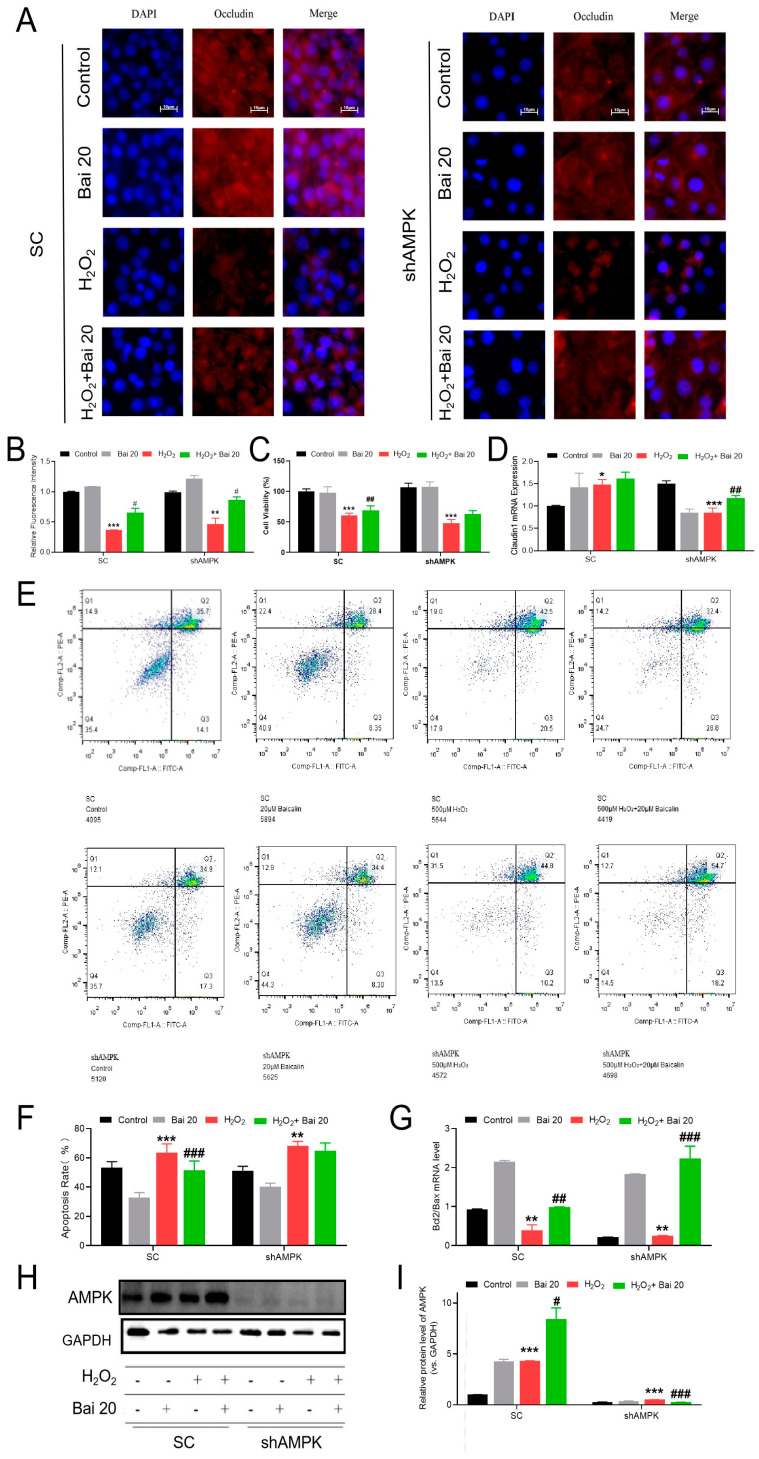
Effects of Bai on the H_2_O_2_-induced intestinal barrier and apoptosis in AMPK-knockdown IPEC-J2 cells. The AMPK-knockdown (shAMPK-IPEC-J2) and/or control (SC-IPEC-J2) cells were pretreated with or without 20 μM Bai for 12 h and then cocultured with 500 μM H_2_O_2_ for 4 h. (**A**,**B**) Immunofluorescence was observed in the sections stained for Occludin under different groups and conditions. Scale bar, 10 μm. (**C**) CCK8 reagent measured the viability of IPEC-J2 cells. (**D**) The mRNA levels of Claudin1 gene expression. (**E**,**F**) Apoptotic cells were analyzed by flow cytometry; results are presented as the percentage of cell viability; values are the mean ± SD. (**G**) The mRNA levels of Bcl2/Bax genes in IPEC-J2 cells under different treatments. Data are shown as ratios of abundance of target gene transcripts in the treated cells to those in the control cells after normalization to β-actin. (**H**,**I**) Protein levels of AMPK were detected by Western blot with GAPDH as the loading control. “*” indicates a significant difference compared with the control group (* *p* < 0.05, ** *p* < 0.01, and *** *p* < 0.001). “#” indicates a significant difference compared with the H_2_O_2_ group (# *p* < 0.05, ## *p* < 0.01, and ### *p* < 0.001).

**Figure 5 ijms-24-09435-f005:**
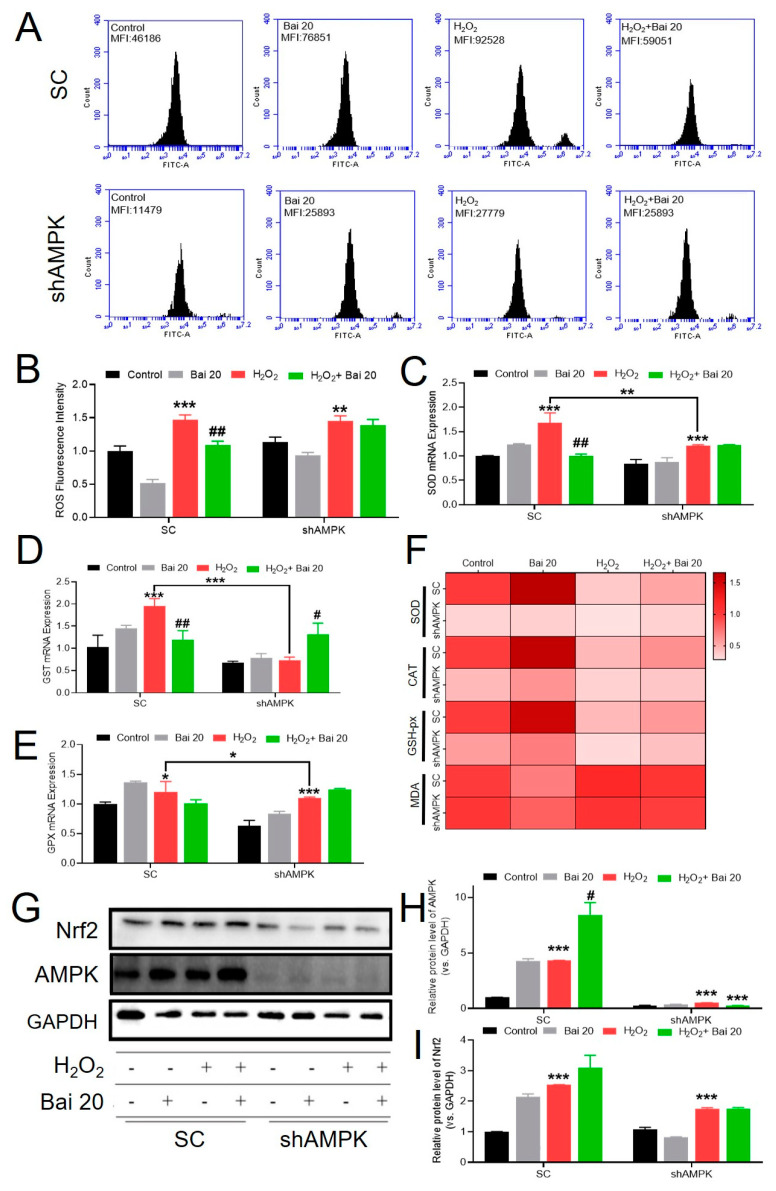
Effects of Bai on H_2_O_2_-induced oxidative stress in AMPK-knockdown IPEC-J2 cells. (**A**,**B**) ROS production; (**C**) SOD activity; (**D**) CAT activity; (**E**) GSH-px activity. (**F**) Heat map showing the expression levels of antioxidant oxidative stress-related mRNA expression in IPEC-J2 cells under different treatments. (**G**–**I**) Protein levels of AMPK and Nrf2 were detected by Western blot with GAPDH as the loading control. “*” indicates a significant difference compared with the control group (* *p* < 0.05, ** *p* < 0.01, and *** *p* < 0.001). “#” indicates a significant difference compared with the H_2_O_2_ group (# *p* < 0.05, ## *p* < 0.01).

**Table 1 ijms-24-09435-t001:** Sequence of hairpin RNA.

Names	Sequence of Hairpin RNA
shAMPK-sense	CCGGTTTCAGGCATCCTCATATAATCTCGAGATTATATGAGGATGCCTGAAATTTTTG
shAMPK-antisense	AATTCAAAAATTTCAGGCATCCTCATATAATCTCGAGATTATATGAGGATGCCTGAAA

**Table 2 ijms-24-09435-t002:** Sequence of target gene primers.

Gene Names	Sequence of Primer (5′–3′)
GAPDH	F:CCTTCCCCTGCGCTCTCT R:TGTAGACCATGTAGTGGAGGTCA F:GCCCTTTTGCTTCAGGGTTTC R:CAATGCGCTTGAGACACTCG F:GGATAACGGAGGCTGGGATG R:TTATGGCCCAGATAGGCACC F:TGTGGGATTGAGACGGACAG R:TCCGTCCTTTGAATTTCGCC F:CACAGCTCCGCTCAGAAC R:GGGACCACCAGTTTGTTCCT F:CTAGTGATGAGGCAGATGAA R:AGATAGGTCCGAAGCAGAT F:GGGTTCTCCTGTCACTGGTAT R:CAGCATGTTTCCGTTTGCCA F:CCTGCTCAGATCCACAATTCC R:GCCAAAGCTTGAGCAGTCTTC F:CCGTGTAACCAGTTCGGACA R:AGCATGAAGTTGGGCTCGAA F:TACCGCTCCCGAATGAACAC R:GTCACGGGAGTGGAGTCTTG F:CAGGTGCACCCTCCAGATTG R:ATGTCGTTGCTGGGTGCATA F:CAGGGCACCATCTACTTCGAG R:CAACGTGCCTCTCTTGATCCT F:TTGATAGTGGCGTTGACA R:CCTCATCTTCATCATCTTCTAC
BAX	
Bcl2	
Caspase3	
Caspase9	
Claudin1	
FAS	
GST	
GPX1	
HO-1	
Occludin	
SOD	
ZO-1	

## Data Availability

The data used to support the findings of this study are available from the corresponding authors upon request.

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
