# Peer review of "Baicalin Attenuates H2O2-Induced Oxidative Stress by Regulating the AMPK/Nrf2 Signaling Pathway in IPEC-J2 Cells"

_ijms, 2023, doi:10.3390/ijms24119435_

Round 1

Reviewer 1 Report

Here are the comments and suggestions for the authors to improve their manuscript.

1、 In FIG1.A, the cell morphology shown in the light picture cannot show their changes upon the changes of intervention. It is suggested to revise this figure for better presentation. “Compared with the control group, the protein and mRNA abundance of ZO-1, Claduin-1, and Occludin were significantly decreased in H2O2 treated group, while pretreating with Bai significantly reversed this downregulation of ZO-1, Occludin, and Claudin1 induced by H2O2(line:94-97), but in FIG 1. E, Bai did not significantly reverse this downregulation mRNA abundance of ZO-1, Claduin-1, Occludin. Moreover, the changes in claduin1 and ZO-1 protein levels were opposite to those mRNA changes, and authors are suggested to clarify such changes. In FIG 1. F , the WB image of Occludin is not clear, and it is recommended to provide high-quality image for this important protein.

2、 In FIG2.Athe apoptosis rate of bai20 group was 0.058%, which was significantly different from the histogram in FIG2.B. Please explain the reason. As shown in Figure 2C, treatment with Bai alone decreased pro-apoptotic relative mRNA expression such as FAS, Bax, Caspase3, and Caspase9, and Bai significant inhibited pro-apoptotic mRNA expression induced by H2O2(P < 0.05 or P < 0.01)”(line: 119-121. Unfortunately, this description was not in line with the results as shown in FIG2.C. For example, bai10 , bai 20 group VS control group is not statistically significant (FAS, Bax, Caspase3), and Caspase9 is increased instead of decreased. Meanwhile, authors are also advised to clarify the inconsistencies between the changes in Caspase3 and Caspase9 protein levels (FIG2.E) and the changes of those mRNA levels (FIG2.C).

3、 3. “while pretreatment with Bai dose-dependent manner significantly increased SOD, CAT and GSH-px activation and decreased MDA activation induced by H2O2 (P < 0.05 or P < 0.001)”line:142-144. In FIG 3C, only two concentrations of 10 uM and 20uM were used, which was inadequately considered to be concentration-dependent, and no significant differences were observed in SOD, CAT, and GSH-px between the two groups. “while pretreatment with Bai significantly reversed this upregulation of HO-1, SOD, and GPX1 induced by H2O2 (P < 0.001).”line: 147-148), but in FIG3.D-E, there is no reversal of HO-1 upregulation by Bai, and the statistical differences between the two antioxidant indexes of SOD and GPX1 (bai+H2O2 vs H2O2) was not P < 0.001.; FIG3.F-G, the decline of p-Nrf2/ Nrf2 after H2O2 treatment can be reversed after bai treatment, which may be the result of the entry of p-Nrf2, the activator of Nrf2, into the nucleus. The manuscript does not specify where the Nrf2 protein (an important antioxidant regulatory point) comes from. From cytoplasm, nucleus, or entirety? Therefore, extraction of protein from the nucleus is suggested to detect p-Nrf2 for confirmation.

4、 In FIG 4Ethe values in the figure are inconsistent with the statistical results in FIG 4F, please check carefully. FIG 4G, “the mRNA level of H2O2+ BAI20 group showed that although bcl2/Bax was significantly increased in AMPK-knockdown IPEC-J2 cells, the apoptosis of the group was still significantly increased, which was intended to indicate that bai could regulate AMPK to inhibit IPEC-J2 cells induced by H2O2, and had nothing to do with the effect of anti-apoptotic gene bcl2/Bax.” Is there any specific reason for not selecting pro-apoptotic genes or other anti-apoptotic genes? Are these results also verified by WB?

5、 In FIG.5, legend: “Protein levels of AMPK, Nrf2 were detected by Western blot with β-actin as the loading control(lines199-200)”.but in FIG.5(G-I) is GAPDHplease reconfirm whether it is β-actin or GAPDH. “The mRNA expression and heatmap analysis of antioxidant genes analysis showed that knockdown of AMPK increased antioxidant genes expression induced by H2O2 (P < 0.05)(line:188-190). It should be “decreased” but not “increased”and no statistical comparison was made between the two groups.

6. “our results showed that Bai significantly attenuated H2O2induced IPEC-J2 cell injury which could increase cell viability, upregulation of the antioxidant enzyme’s ability, regulate the relative of intestine barrier mRNA and protein levels.”(line211-213) Intestine barrier mRNA level is not statistically significant and the summary here is not precise.

7. and then sharply decreases ROS production and alleviating mitochondria mediated apoptosis.”(line:257-258). Relevant experiments for mitochondria mediated apoptosis were not provided. Please clarified whether such experiments were performed in this manuscript.

8.In present study, our result showed that Bai could protect against oxidative stress-induced damage by elevating sestrin 2 (SESN2) and kelch like ECH-associated protein 1 (Keap1) expression of AMPK signaling pathwaylines287-289”. Relevant experiments for SESN2 and Keap1 were not provided. Please clarified whether such experiments were performed in this manuscript.

Author Response

Dear editors and reviewers,

On behalf of my co-authors, we appreciate editors and reviewers very much for their positive and constructive comments and suggestions on our manuscript entitled “Baicalin attenuate H2O2-induced oxidative stress by regulating AMPK/Nrf2 signaling pathway in IPEC-J2 cells”.

We have studied the reviewer’s comments carefully and have made revisions which are marked in yellow highlighting in the paper. We have tried our best to revise our manuscript according to the comments. Attached please find the revised version, which we would like to submit for your kind consideration.

We would like to express our great appreciation to you and reviewers for comments on our paper. Looking forward to hearing from you.

Thank you and best regards.

Yours sincerely,

Jiahua Liang

Corresponding author: Guoliang Hu

Reviewer 2 Report

[General]

The aim of this manuscript is to clarify the effects of baicalin on H2O2-induced oxidative stress and AMPK/Nrf2 pathway in intestinal epithelial cells. The authors showed that baicalin improved H2O2-induced apoptosis and ROS production, and that the effects of baicalin were abolished by AMPK knockdown. These results in this study suggest that baicalin improve oxidative stress-induced injury and apoptosis of intestinal cells. However, the descriptions contain many discrepancy among Abstract, Results and Discussion in this manuscript. Therefore, many correction that should be checked before submission are needed.

[Major points]

1. The authors concluded that baicalin attenuate H2O2-induced oxidative stress by regulating AMPK/Nrf signaling pathway. However, the authors showed that baicalin alone affected several molecules. For example, baicalin up-regulated the expression of occludin , the enzyme activity of SOD, CAT and GST. Also, baicalin down-regulated apoptosis, the production of ROS, the expression of Fas, Bax and caspase-3. I consider that these results do not necessarily show that baicalin affects H2O2-induced these changes.

2. The description in Abstract did not match to those in Results. The sentence "Bai up-regulated the AMPK phosphorylation and Nrf2 phosphorylation level" (Page 1, line 24) is apparently distinct to Fig. 3F and 3G.

3. The population with high fluorescent intensity of ROS is present in H2O2-stimulated cells and this population is disappeared by the pretreatment of baicalin in Fig 5A (SC). In contrast, this population is not detected in Fig 3A. The possibility is considered that the same population is present but is not detected due to sensitivity of X-axis.

4. The authors show the expression of Nrf2 protein in Fig 5G. The results of phosphorylated Nrf2 should be showed as well as in Fig 3F-G because phosphorylation of Nrf2 is important for its activity. Related to this, the authors described "the phosphorylation level of Nrf2 was affected when AMPK was knocked down" (Page 11, line 303-304).

5. The authors described that TJ proteins expression are "in dose-dependent manner" (Page 10, line 252). However, such data is not presented in this manuscript. This description should be corrected.

[Minor points]

1. The compositions among figures are inconsistent. For example, the order of ZO-1/occuludin/claudin1 is different between Fig 1E and 1G. In addition, GAPDH is present at the top in Fig 1F and Fig 2D, whereas this is at the bottom in Fig 3F and Fig 4G. These points should be improved for readers.

2. The author should show the results of AMPK protein expression to confirm the effect of AMPK-knockdown by shRNA introduction in Fig 4 as well as in Fig 5G.

3. Figure legends of Fig 3F and 3G is inappropriate and should be corrected.

4. All antibodies should be described in Materials and Methods. The description about antibodies against AMPK/p-AMPK/Nrf2/p-Nrf2/GAPDH and secondary antibodies are missing.

5. Violin plot is used in Fig 5B. However, violin plot is less meaningful when sample size is small. Bar chart with SD bar is sufficient in this case.

6. The legend of Fig 5G is wrong. β-actin is not used in this experiment. The authors should be checked before the submission.

7. typo?: plate not plant (Page 12, line 326) and 500 μM not 500 mM (Page 12, line 328).

Author Response

(The authors gave the same response as above.)

Round 2

Reviewer 1 Report

1“Pretreating with Bai reversed this downregulation of Occludin induced by H2O2 (P < 0.05).”line96-97. In FIG.E, however, your result of PCR for Occludin indicated no significant difference compared with H2O2 groupbai+H2O2 VS H2O2. Moreover, in FIG 1. F , the WB of Occludin is not clearwhich is not adequate to support your conclusion.

2In FIG2.Athe apoptosis rate of bai20 group was 0.058%, which was significantly different from the histogram as stated in FIG2.B. Although the bar chart has been corrected, it is recommended that a more representative picture, inFIG2A, should be replaced.

3In the third item of the last review opinion, the authors did not explain or clarify the abnormal result of p-Nrf2/ Nrf2 protein, which is difficult to understand. Similarly, the review comment No. 7 was not replied.

Author Response

(The authors gave the same response as above.)

Reviewer 2 Report

The authors answered my questions and corrected manuscript.

Please correct typo in Figure legends: H2O2 (line 105, 113, and 176) to H2O2, and mean +- S (line 131 and 180) to mean +- SD.

Author Response

(The authors gave the same response as above.)
